# A Hierarchical Model for Device Placement

**Azalia Mirhoseini\*, Anna Goldie**\*, **Hieu Pham, Benoit Steiner, Quoc V. Le and Jeff Dean**
`{azalia,agoldie,hyhieu,bsteiner,qvl,jeff}@google.com`

## Abstract

We introduce a hierarchical model for efficient placement of computational graphs onto hardware devices, especially in heterogeneous environments with a mixture of CPUs, GPUs, and other computational devices. Our method learns to assign graph operations to groups and to allocate those groups to available devices. The grouping and device allocations are learned jointly. The proposed method is trained with policy gradient and requires no human intervention. Experiments with widely-used computer vision and natural language models show that our algorithm can find optimized, non-trivial placements for TensorFlow computational graphs with over 80,000 operations. In addition, our approach outperforms placements by human experts as well as a previous state-of-the-art placement method based on deep reinforcement learning. Our method achieves runtime reductions of up to 60.6% per training step when applied to models such as Neural Machine Translation.

## 1 Introduction & Related Work

Deep neural networks have been successfully applied to many practical problems, such as image classification (LeCun et al., 1998; Krizhevsky et al., 2012; Taigman et al., 2014; Szegedy et al., 2015), speech recognition (Hinton et al., 2012; Hannun et al., 2014), and machine translation (Sutskever et al., 2014; Bahdanau et al., 2015; Wu et al., 2016b). These successes have lead to a surge in demand for the computational resources needed to train and infer with neural networks. A common approach to addressing this demand is to use a distributed environment with a combination of CPUs and GPUs. In this environment, it is typical for a machine learning practitioner to explicitly place the operations of their neural network onto particular computing devices for model parallelism and data parallelism. For example, one might distribute the computation of the first layer in a translation network onto the first GPU and the computation of the second layer onto the second GPU (Sutskever et al., 2014; Wu et al., 2016b). Although these decisions can be made by a human practitioner, such an approach does not scale well or produce optimal results, especially in the case of more complicated networks (Szegedy et al., 2016b;a). Given the growing diversity of hardware devices (e.g., Google TPUs, Intel Nervana, etc.) and recent trends toward automated neural architecture search (Zoph & Le, 2017; Real et al., 2017; Baker et al., 2016), where new models are generated, trained and evaluated in an entirely end-to-end fashion, it seems natural to move toward more automated solutions for efficiently distributing computation.

Device placement can be framed as the problem of learning to partition a graph across available devices. Given that graph partitioning is a well-studied subject in computer science (Fiduccia & Mattheyses, 1988; Karypis & Kumar, 1995b; Pellegrini, 2009b), traditional graph partitioning methods represent a natural baseline for automated device placement. We ran experiments using Scotch (Pellegrini, 2009b), a well-established open source library for graph partitioning, which includes optimizations such as k-way Fiduccia-Mattheyses (Fiduccia & Mattheyses, 1988), Multilevel methods (Barnard & Simon, 1994; Hendrickson & Leland, 1993; Karypis & Kumar, 1995a), the Band Method (Chevalier & Pellegrini, 2006), the Diffusion Method (Pellegrini, 2007), and Dual Recursive Bipartitioning Mapping (Pellegrini & Roman, 1996). The objective was to balance the computational load across a set of connected processing nodes, while colocating neighboring nodes to minimize communication cost. Despite its promise, this approach yielded disappointing results, likely due to the non-stationarity of the cost function. We target a distributed environment where we use a shared cluster of CPUs and GPUs, and our CPUs may also serve other jobs at the same time. Thus, while cost-based models such as (Matthias Boehm & Tian, 2014) provide a strong baseline

---

\*Equal Contribution.

for memory optimizations, since memory usage is deterministic, they cannot be directly applied to environments with dynamic costs.

Using deep networks and reinforcement learning for combinatorial optimization has already been proposed (Vinyals et al., 2015; Bello et al., 2016; Mirhoseini et al., 2017). Recent work (Mirhoseini et al., 2017) uses a recurrent neural network (RNN) policy network to predict the placement of operations in a computational graph, optimizing for speed of computation using policy gradient methods. While this approach outperforms traditional graph partitioning heuristics and human expert placements, it is prohibitively expensive for the RNN policy to learn when the number of operations is large. This method is therefore limited to small graphs (with fewer than 1000 nodes) and requires human experts to manually partition the graph into collocation groups as a pre-processing step in order to scale to larger graphs. We refer to the method in (Mirhoseini et al., 2017) as ColocRL.

In this paper, we propose a more flexible approach which learns to optimize device placement for training neural networks that have tens of thousands of operations with no need for manual grouping. Our method consists of a two-level hierarchical network, in which the first model groups the operations of the graph (the Grouper) and the second model places those groups onto devices (the Placer). The Grouper is a feed forward network which reads in information about each operation and its context within the graph, in order to predict the group to which that operation should be assigned. The Placer is a sequence-to-sequence model (Sutskever et al., 2014) that reads in the embedding of the group and predicts the device placement for that group. The entire two-level network is trained jointly using reinforcement learning to optimize for speed of computation and for feasibility (e.g., having sufficient memory available on each device for the computation assigned). Unlike the previous work, our method is end-to-end and does not require human experts to manually group operations as a pre-processing step, making it a fully automated solution to optimizing device placement.

Our main result is that our model effectively handles very large graphs and finds non-trivial placements on multiple devices for models such as Inception-V3 (Szegedy et al., 2016b), ResNet (He et al., 2016), Language Modeling (Jozefowicz et al., 2016), and Neural Machine Translation (Wu et al., 2016b). The placements found by our model outperform TensorFlow's default placements (Abadi et al., 2016), the Scotch algorithm's placements, and human expert placements, as well as those of ColocRL (Mirhoseini et al., 2017). Our results demonstrate that the proposed approach learns the properties of the environment, including the complex tradeoff between computation and communication in hardware. For example, on a Neural Machine Translation model, our method achieves a 60.6% reduction in training time per iteration.

## 2 METHOD

An overview of our hierarchical model for device placement is shown in Figure 1. Our model consists of two sub-networks: A Grouper that assigns operations to groups and a Placer that assigns groups to target devices. The two models are trained jointly.

The objective of the proposed approach, which we refer to as the Hierarchical Planner, is to predict a placement that speeds up the training of neural network graphs. The runtime we are optimizing for is the time taken to conduct one forward pass, one back-propagation pass, and one parameter update on the target neural network. To measure the runtime, the predicted placement is run on actual hardware. Since the reward (runtime) in this problem is non-differentiable, we use policy gradients to train the Hierarchical Planner. Moreover, the policy gradients flow through to train both the feed forward Grouper and the recurrent Placer.

The Grouper assigns each operation to a group. Once all the operations are grouped, we use information about each member operation to generate an embedding for that group. We then pass these embeddings as input to the Placer, which computes device placements for each group. The Placer assigns zero or more groups to each available device. The final placement is determined by placing each operation on the device that its group was assigned to.

In our implementation, the Grouper is a feed forward model followed by a softmax layer with an output size equal to the number of groups. The Placer is a sequence-to-sequence model (Sutskever et al., 2014) with Long Short-Term Memory (Hochreiter & Schmidhuber, 1997) and a content-based attention mechanism (Bahdanau et al., 2015) to predict the placements.

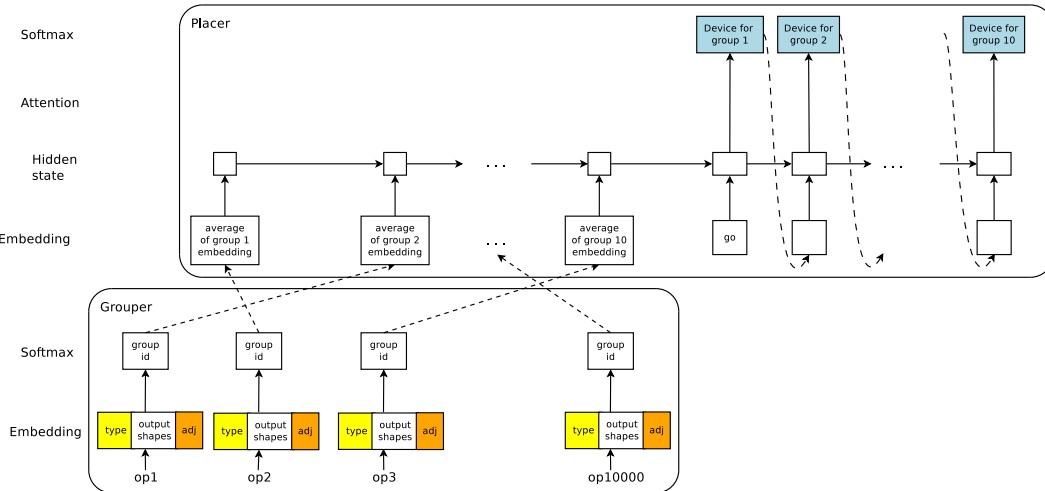

Figure 1: Hierarchical model for device placement (see text for more details).

We first generate operation embeddings to pass as input to the Grouper. Each operation embedding consists of 3 vectors: 1) A vector that embeds operation type information (e.g., MatMul, Conv2d, Sum, etc.). We treat this as a language modeling task and learn an operation type embedding of size 20 with a vocabulary of the 200 most commonly used TF operations. 2) A vector that contains output sizes and number of outputs for each operation. We limit the number of output edges to 6 and the size of each of these outputs to 4 elements. We populate this vector by reading the outputs of an operation one by one and inserting the output operations shapes. We pad the vector with -1 if we have fewer outgoing edges or smaller sizes. 3) A vector that contains adjacency information for that operation. We index the graph by traversing it in a BFS manner and set the maximum number of incoming and outgoing edges to 12 (6 for each direction). We then fill the vector with the index of the incoming and outgoing operations. We pad the vector with -1, in cases where the number of incoming or outgoing edges is less than 6.

To generate input for the Placer, we take each group and create its group embedding by concatenating 3 vectors: 1) A vector containing the count of each operation type in the group. 2) A vector that counts the total number of output shapes of all the operations in that group. This vector is created by concatenating all the operation output shape embeddings described above (not including the -1) and is of size 16. 3) A vector that contains group adjacency information. The size of this vector is the number of groups (256 in our experiments), and its i-th value is 1 if the group has edges to the i-th group and 0 otherwise.

The Placer's RNN encoder reads the group embeddings one at a time and produces $M$ hidden states. We treat $M$, which is equal to the number of groups, as a hyperparameter. The Placer's decoder RNN predicts one device per time step. The devices are returned in the same order as the input group embeddings, i.e., the operations in the first group will be placed on the device returned by the first decoder step, and so on. Each device has its own trainable embedding, which is then fed as input to the next decoder time step.

At each step $t$ (where $1 \leq t \leq M$), the decoder uses an attention mechanism to attend over the encoder states. We use the attention mechanism from Vinyals et al. (2015). At training time, the decoder samples one device $d_t$ per step from the Placer's softmax. To make the activations $l_t$ less steep and to allow the model to explore, we follow Bello et al. (2016) and use a temperature $T$ and apply a $\tanh$ constant $C$ to $l_t$. Thus, we sample $d_t$ as follows:

$$d_t \sim \text{softmax}(C \tanh(l_t/T)) \tag{1}$$

The placement decisions are then used to place the model. In the following section, we describe a policy gradient method to train the Hierarchical Planner, such that it improves its decisions over time.

**Training with REINFORCE:** The planner optimizes the training time for a target model (e.g., a TensorFlow graph) given the decisions made by the Grouper and the Placer. Let $r_d$ be the runtime

per training step for a predicted device placement $d$. We define the reward for placement $d$ as $R_d = -sqrt(r)$. The planner should try to maximize the expectation of $R_d$ given its decisions. As such, the cost function we are optimizing for is:

$$J(\theta_g, \theta_d) = \mathbf{E}_{\mathbf{P}(\mathbf{d};\theta_\mathbf{g},\theta_\mathbf{d})}[R_d] = \sum_{g \sim \pi_g} \sum_{d \sim \pi_d} p(g;\theta_g)p(d|g;\theta_d)R_d \tag{2}$$

Let $\theta_g$ and $\theta_d$ be parameters of the Grouper and Placer, respectively. Here, $p(g;\theta_g)$ is the probability of a sample group assignment $g$ drawn from the Grouper softmax distribution $\sim \pi_g$ and $p(d;\theta_d)$ is the probability of a sample device placement $d$ drawn from the Placer softmax distribution $\sim \pi_d$. We can write the derivative of the cost function defined in Eq. 2 w.r.t. $\theta_g$ and $\theta_d$ as follows:

$$\nabla_{\theta g} J(\theta_g, \theta_d) = \sum_{g \sim \pi_g} \nabla_{\theta g} p(g;\theta_g) \sum_{d \sim \pi_d} p(d|g;\theta_d)R_d \tag{3}$$

$$\approx \frac{1}{m} \sum_{g_i \sim \pi_g}^{1 \le i \le m} \nabla_{\theta g} \log p(g_i;\theta_g) . \frac{1}{k} (\sum_{d_j \sim \pi_d}^{1 \le j \le k} R_{d_j}) \tag{4}$$

$$\nabla_{\theta d} J(\theta_g, \theta_d) = \sum_{d \sim \pi_d} \sum_{g \sim \pi_g} p(g;\theta_g) \nabla_{\theta d} p(d|g;\theta_d)R_d \tag{5}$$

$$\approx \frac{1}{k} \sum_{d_j \sim \pi_d}^{1 \le j \le k} \frac{1}{m} (\sum_{g_i \sim \pi_g}^{1 \le i \le m} \nabla_{\theta d} \log p(d_j|g_i;\theta_d)R_{d_j}) \tag{6}$$

Deriving Eqs. 3 and 5 from the cost function is straightforward. We use the REINFORCE rule (Williams, 1992) and approximate expectation values with samples $g_i$ and $d_j$ drawn from the Grouper and Placer to arrive at Eqs. 4 and 6.

In our implementation, the Grouper makes independent predictions when assigning operations to groups. The Placer, however, conditions the placement of groups on those that have already been placed. To reduce the variance, we also subtract a baseline $B$ from $R$. In our experiments, we found that the exponential moving average of the reward was an effective baseline.

**Distributed Training:** Our policy is trained in a distributed manner. Our framework has a parameter server that is shared among several controllers. All controllers use the same set of parameters and update the policy asynchronously. Each controller communicates with $k$ worker nodes, where $k$ is as shown in Eqs. 4 and 6. Each worker interacts with only one controller.

Each worker executes the placement given by its controller and reports the runtime. In our experiments, we use 4 controllers and 16 workers (4 per controller). For example, if we are optimizing a placement on 2 GPUs, each worker needs 2 GPUs to measure runtime. Each controller is hosted on a single GPU. Therefore, we use a total of 36 GPUs for this example. The workers run the placements in parallel. Once all workers have finished running the placements, the controller computes the gradients using the measured runtimes. To reduce the variance of runtime measurements across workers (different machines), each controller maintains a separate baseline. While we get our best results by using many workers, we show in Section 3 that it is possible to train the policy and achieve comparable results with far fewer resources.

## 3 EXPERIMENTS

In this section, we apply Hierarchical Planner to widely used machine learning models in computer vision and natural language processing. We compare our results to heuristic and RL-based graph optimization baselines and demonstrate our approach's ability to find performant placements. We also compare our method with two simpler alternatives: 1). no grouping (a feed forward model that directly places each operation) and 2). random grouping (a random Grouper feeding into a learned Placer), to demonstrate that our hierarchical architecture allows us to learn better placements.

**Models:** We evaluate our approach on four widely used deep neural networks:

- **Inception-V3** (Szegedy et al., 2016b) is a model used for a variety of computer vision tasks, including classification, recognition, or generation (Khetan & Oh, 2016; Esteva et al., 2016). The network consists of multiple blocks, each of which is made up of several convolutional and pooling layers. Within a block, the layers can be executed in parallel. However, since the outputs of each block are concatenated together to form the input to the next block, the blocks must be executed sequentially. We use a batch size of 1. The TensorFlow graph encoding this model contains 24,713 operations.

- **ResNet** (He et al., 2016) is a popular model for image classification. It is a deep convolutional network that uses residual connections to avoid the vanishing gradient problem. We use batch size 128. The TensorFlow implementation of this model has 20,586 operations.

- **RNNLM** (Zaremba et al., 2014; Jozefowicz et al., 2016) Recurrent Neural Network Language Model is made of many LSTM cells organized in a grid structure. The processing of each LSTM cell only depends on the results of 2 other cells, which make the concurrent execution of many LSTM cells possible given enough hardware resources. We use batch size 64. The corresponding TensorFlow graph contains 9,021 operations.

- **NMT** (Bahdanau et al., 2015; Wu et al., 2016a) Neural Machine Translation with attention mechanism has an architecture similar to that of RNNLM, but its many hidden states make it far more computationally expensive. To decrease the training time, both Sutskever et al. (2014) and Wu et al. (2016a) propose placing each LSTM layer, as well as the attention and the softmax layer, on a separate device. While this strategy results in meaningful speed improvements, we show that our Hierarchical Planner can find significantly better placements. We use batch size 64. We evaluated 3 versions of the NMT model:
  - The original 2-layer encoder-decoder consisting of 28,044 operations.
  - An extended 4-layer version consisting of 46,600 operations.
  - An even larger 8-layer version consisting of 83,712 operations.

For a fair comparison to previous state-of-the-art deep RL methods (Mirhoseini et al., 2017), we use the same model architectures (Inception-V3, RNNLM, and 2-Layer NMT models), hyperparameters and input data. In addition, we evaluate our model on a 152-layer ResNet (He et al., 2016) with ImageNet data (Deng et al., 2009), as well as more complex NMT models with 4 and 8 layers.

**Baselines:** We compare the placements found by our approach to the following baselines:

- **CPU Only.** Here, we execute the model on a single CPU. While this is generally slow, we find that some large models are very hard to fit on GPUs, given their memory limitations, leaving a CPU-only placement as the only naive option.

- **GPU Only.** In cases where it is possible to fit the entire model on a single GPU, this is a strong baseline as most graph operations run fastest on GPU and this placement incurs no cross-device communication cost. For operations that are not implemented on GPU, TensorFlow automatically places them on CPU.

- **Scotch.** We use the Scotch static mapper (Pellegrini, 2009a) which takes as input the graph, the computational cost of each operation, the volume of data flowing through each edge, and the compute and communication capacities of the pertinent devices.

- **MinCut.** We use the Scotch optimizer, but we only consider GPUs as our devices. The objective is to balance computation across all the available devices while minimizing inter-device communication.

- **Human Expert.** We use hand-crafted placements from previous publications. For Inception-V3 (Szegedy et al., 2016b) and Resnet (He et al., 2016), where it is difficult to exploit model parallelism, human experts place the graph on a single GPU. For RNNLM and NMT, existing work (Sutskever et al., 2014; Wu et al., 2016a) places each LSTM layer on a separate GPU. For NMT, the attention and softmax layers are placed on the same device as the final LSTM layer, while the embedding layer is colocated with the first LSTM layer.

- **ColocRL.** This method (Mirhoseini et al., 2017) uses policy gradient to train a recurrent neural network that reads in hand-crafted colocation groups and then places each group on a device.

**Measuring Reward:** Our reward is the negative square root of the runtime for one training step of the target TensorFlow model (lower runtimes are better). We assign a large negative reward of -10 to invalid placements (e.g., due to memory limitations). We define runtime as the time in seconds required to complete one training step of the target model (i.e. one forward pass, one back-propagation pass, and one parameter update). To reduce measurement variance, we run each predicted placement of the model for 10 steps. We discard the first 5 steps (to avoid warmup variations) and use the median value of the next 5 steps to calculate the reward. We found empirically that calculating the reward with square root yielded better placements than identity or logarithm. Note that by altering the reward, we can use our proposed method for optimizing other metrics, such as inference speed, throughput, and network congestion.

**Devices and Software:** Our experiments are run on machines with 1 Intel Haswell 2300 CPU and up to 8 Nvidia Tesla K40 GPUs. We use TensorFlow r1.3 to run our experiments.

**Architecture of the Policy Network:** In the Hierarchical Planner, the Grouper is a feed forward network with a hidden size of 64 and the Placer is a sequence-to-sequence (Sutskever et al., 2014) model with an LSTM hidden size of 256. For the encoder of the sequence-to-sequence model, we used two layers of LSTM to form a bi-LSTM similar to (Wu et al., 2016b). We used a uni-directional LSTM for the decoder. The Grouper's softmax output size is equal to the number of groups, which we set to 256 in our experiments. We also experimented with a range of group sizes (64 to 1024), but got the best results with group size 256. The number of unrolled steps in the Placer is equal to the number of groups. The Placer's softmax output size in both models is equal to the number of available hardware devices.

**Training Details:** We train both policies using Adam (Kingma & Ba, 2015) optimizer with a fixed learning rate of 0.1, gradient clipping of norm 1.0, tanh constant $C = 5.0$, and temperature $T = 10.0$. The number of Grouper and Placer samples in Eqs. 4 and 6 are $m = 1$ and $k = 4$, respectively.

To encourage more exploration, we added noise to the logits of both the Grouper and the Placer networks for the first 500 policy training steps. The noise was sampled from the normal distribution and modulated to have a max amplitude of 0.1.

Given that the vast majority of placements are invalid, especially for more complex models such as NMT, we update the policy only for valid placements after the first 500 steps. By updating the baseline and the policy only for samples that give valid placements, we prevent the policy from converging to the reward associated with invalid placements.

| Tasks | CPU Only | GPU Only | #GPUs | Human Expert | Scotch | MinCut | Hierarchical Planner | Runtime Reduction |
|---|---|---|---|---|---|---|---|---|
| Inception-V3 | 0.61 | 0.15 | 2 | 0.15 | 0.93 | 0.82 | **0.13** | 16.3% |
| ResNet | - | 1.18 | 2 | 1.18 | 6.27 | 2.92 | **1.18** | 0% |
| RNNLM | 6.89 | 1.57 | 2 | 1.57 | 5.62 | 5.21 | **1.57** | 0% |
| NMT (2-layer) | 6.46 | OOM | 2 | 2.13 | 3.21 | 5.34 | **0.84** | 60.6% |
| NMT (4-layer) | 10.68 | OOM | 4 | 3.64 | 11.18 | 11.63 | **1.69** | 53.7% |
| NMT (8-layer) | 11.52 | OOM | 8 | **3.88** | 17.85 | 19.01 | 4.07 | -4.9% |

Table 1: Model Runtimes (seconds) for different placements (lower is better). OOM: Out Of Memory.

**Results Compared with Graph Partitioning Heuristics:** In Table 1, we report the performance of the Hierarchical Planner and compare it to the aforementioned baselines. The only information available to our models is the TensorFlow graph and a list of devices. The reduction percentages are computed by taking the difference between the runtime achieved by the Hierarchical Planner and that of the best prior placement, and then dividing it by that best prior runtime.

For ResNet and RNNLM, our model learns that it is more efficient to use a single GPU, as this minimizes communication cost. For Inception-V3, the Hierarchical Planner learns to distribute the model across 2 GPUs, achieving a 16.3% reduction in runtime over placing the model on a single GPU.

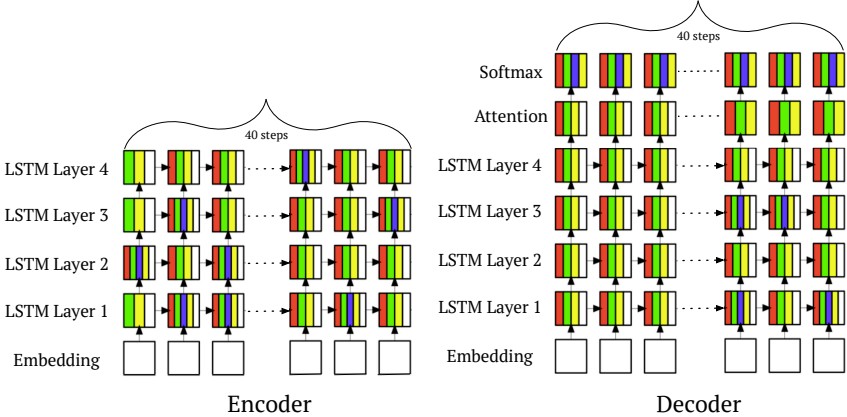

Figure 2: The Hierarchical Planner's placement of a NMT (4-layer) model. White denotes CPU and the four colors each represent one of the GPUs. Note that every step of every layer is allocated across multiple GPUs. This placement is $53.7\%$ faster than that generated by a human expert.

For NMT with 2, 4, and 8 layers, we ran experiments with 2, 4, and 8 GPUs, respectively. We outperform the best prior results by $60.6\%$ for NMT (2-layer) and $53.7\%$ for NMT (4-layer). For further insight into the model's behavior, we visualize its placement for NMT (4-layer) in Figure 2.

For NMT (8-layer), the Hierarchical Planner finds a placement that is $4.9\%$ slower than that of human experts. Even in this one case where the method slightly underperforms, it is still useful to have an automated method of finding placements that are comparable to those of human experts.

Results associated with both Scotch and MinCut were significantly worse than human expert baselines, which is consistent with results reported in (Mirhoseini et al., 2017).

**Results Compared with ColocRL:** A fair comparison with ColocRL would require us to run the models using exactly the same software (TensorFlow version) and hardware (CPU and GPU types). Although our runtimes are considerably faster, they are not directly comparable to those reported in (Mirhoseini et al., 2017), because we ran with different GPU types (the slower k40 for us vs. their k80) and TensorFlow versions (our r1.3 vs. their unspecified but presumably earlier version). We will discuss the relative improvements achieved by our approach.

For NMT (2-layer), our improvement over best heuristics is $60.6\%$, compared to $19.0\%$ for ColocRL. For NMT (4-layer) and NMT (8-layer), no results were reported for ColocRL, which we suspect is due to the model being unable to handle the large number of operations in these graphs.

Unlike our method, ColocRL makes the strong assumption that certain operations must be colocated. Figure 2 shows the high granularity of the Hierarchical Planner's placements, a degree of parallelism that would be infeasible for prior methods. For example, the Hierarchical Planner places each step of an unrolled LSTM layer across multiple GPUs, whereas ColocRL colocates all operations in a step.

**Analysis:** Here, we want to understand and analyze the placements generated by the RL model. In Figure 2, we show a portion of the placement found by the Hierarchical Planner for NMT (4-layer). With this placement, the runtime per training iteration is $53.7\%$ faster than that of a hand-crafted placement. As shown by the figure, the generated placement is non-trivial and highly parallelized. In particular, all of the unrolled steps of the LSTM, attention, and softmax layers are distributed across multiple GPUs. Note that it is impossible for an approach like ColocRL (Mirhoseini et al., 2017) to find such a placement, as this method forces all operations within an unrolled LSTM step to be placed on the same device. Our method also learns to place the sparse embedding lookup operations on CPU. In this placement, the policy search space is incredibly large, i.e., $5^{46,600}$ (5 devices and 46,600 operations). The automated placement enabled by jointly learned grouping not only outperforms previous methods, but unlike ColocRL, it is deployable with no human effort (e.g. manual grouping).

In our experiments, we set the number of groups to 256. While training the policy, we observed that initially the operations are assigned nearly uniformly across all the 256 groups, but that the Grouper

ultimately converges to using only a small subset of groups across all models. This suggests that the feed forward Grouper has learned to partition the computational graph such that operations that should be placed on the same device are grouped together.

We cast the device placement problem as a sequential decision-making task. Since there is no canonical order for a TensorFlow graph, we randomized the order of the groups that we fed into the the Placer, with the Placer's bi-LSTM architecture enabling it to look at a graph more holistically. We ran 10 experiments on our NMT (4-layer) baseline, shuffling the order of group embeddings passed to the Placer. The difference between the fastest and slowest placements was less than 7%.

**Overhead of Training Hierarchical Planner:**   For each of the models, we train a new policy which learns to optimize placements for that particular model. All results for the Hierarchical Planner are after 1000 iterations of updating the policy. In practice, this takes at most three hours for our largest benchmark. The runtime per policy update is dominated by the time it takes to measure the reward for a sampled placement. To calculate the reward, we run the target model according to the predicted placement for 5 training steps and use the median runtime. The policy itself is a lightweight network that is trained on a single GPU. For measuring the reward, however, we use actual runtime measurements for the given placements. Measuring runtime is done by worker nodes. For example, if we are optimizing a model placement on 5 devices (1 CPU and 4 GPUs), we need at least one worker with that many devices that can run the input model for the predicted placements and report the runtime.

As described in Section 2, we used 16 workers to train our policy. However, in this section, we will show that we can generate good results even in limited hardware settings. We consider training the policy to optimize placement of our 4-layer NMT benchmark model on 5 devices (1 Intel Haswell 2300 and 4 Nvidia Tesla K40s). The goal is to show that the policy can be trained efficiently even when we have only one worker. Figure 3 demonstrates the policy loss reduction as a function of policy training time, in 2 different scenarios where we have access to 1 and 4 workers. The policy is hosted on a single K40 GPU which sends 1 placement at a time to the worker(s) and applies a gradient step for each reward collected from worker(s). While, more workers can reduce the policy training time, as seen in Figure 3, we still get reasonable training times with only one worker. In this case, it takes less than 2.5 hours for the policy to achieve a placement with training step time of 1.94 seconds.

Given that we train NMT models for hundreds of thousands of steps, the overhead of policy optimization is more than justified. For example, to train WMT'14 En->Fr dataset which has more than 36 million examples for one epoch (with batch-size=64), we need to run the NMT model for approximately 562500 steps. Since we reduce the runtime per step by approximately 46.7% (from 3.64 to 1.94 seconds), this saves us 265 GPU-hours, which is a significant savings even if we consider the 12.5 GPU-hours we spent on training the policy on a single worker.

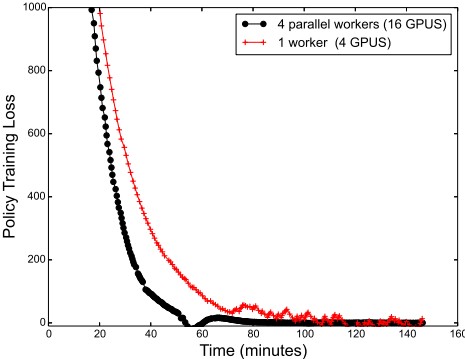

Figure 3: Training the policy with 1 and 4 workers to measure the reward. Each worker is a platform with 1 Intel Haswell 2300 and 4 Nvidia Tesla K40s.

For simplicity, we first train the Hierarchical Planner and then use its best placement to train the target model. For greater efficiency, however, we could interleave the training of the Hierarchical Planner with the training of the target model, using the runtime of actual training steps as our reward signal.

**Alternative Policy Architectures:** We compared the Hierarchical Planner against two alternative policy architectures described below.

The first alternative we considered was a simpler model consisting of a single feed forward network. This model, which we refer to as the Simple Planner, independently predicts the placement for each operation in the input model, given information about that operation and its connectivity to others. This is equivalent to having only a feed forward Grouper which predicts placements rather than groups (i.e. the number of groups is equal to number of available devices).

The Simple Planner finds placements for Inception-V3 that are within 20% of the Hierarchical Planner, and it successfully learns to place RNNLM and ResNet benchmarks on a single GPU. Its learned placement for NMT (2-layer), however, is more than twice as slow as that of the Hierarchical Planner. The Simple Planner also fails to find any valid placements for larger benchmarks, such as NMT with 4 or 8 layers. The Hierarchical Planner which breaks the problem into grouping and placing sub-tasks is able to scale to much larger models. The Hierarchical Planner's sequence-to-sequence model also enables conditioning placement of an operation on those previously placed.

Another architecture we considered was a Hierarchical Planner with randomized grouping. To verify that the Grouper was contributing meaningfully to the performance of the Hierarchical Planner, we compared its performance to a baseline where we fed randomized groupings into the Placer. We ran this experiment with 10 different randomized group assignments for 1000 iterations. As can be seen in Table 2, there is significant variance across different trials and the best result is worse than that of the Hierarchical Planner. This suggests that end-to-end learning of grouping operations and placing groups does indeed improve the performance.

| Benchmark | Best | Median | Worst | Improvement with Hierarchical Planner |
|---|---|---|---|---|
| Inception-V3 | 0.22 | 0.51 | 0.65 | 40.9% |
| ResNet | 1.18 | 1.18 | 1.18 | 0% |
| RNNLM | 1.57 | 1.57 | 1.57 | 0% |
| NMT (2-layer) | 2.25 | 3.72 | 4.45 | 62.7% |
| NMT (4-layer) | 3.20 | 3.42 | 6.91 | 47.2% |
| NMT (8-layer) | 6.35 | 6.86 | 7.23 | 35.9% |

Table 2: Best, median and worst runtimes for 10 trials each with a different randomized grouping of operations. These results demonstrate that the Hierarchical Planner, which uses learned groupings rather than random ones, is able to significantly reduce runtime.

## 4 CONCLUSION

In this paper, we present a hierarchical method for efficiently placing the operations of a computational graph onto devices. Our approach consists of a hierarchical model that first assigns the operations to groups and then places those groups onto devices. We use a policy gradient method to optimize the parameters of the planner. The proposed method enables us to scale to computational graphs containing over 80,000 operations. Unlike previous work, our method is end-to-end and requires no manual effort. On a range of tasks including image classification, language modeling, and machine translation, our method surpasses placements designed by human experts as well as those of previous state-of-the-art deep RL methods. Our approach finds highly granular parallelism within the graph, enabling us to outperform prior methods by up to 60.6%.

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
