# OpenReview forum: "A Hierarchical Model for Device Placement"
_ICLR.cc/2018/Conference — Accept (Poster)_

### Official Review · AnonReviewer3 · 2017-11-22
**The paper proposes to jointly learn groups of operators to colocate and to place learned groups on devices so as to distribute operations for deep learning via reinforcement learning.**

**Rating:** 5
**Confidence:** 4

**Review:**

The paper seems clear enough and original enough. The idea of jointly forming groups of operations to colocate and figure out placement on devices seems to hold merit. Where the paper falls short is motivating the problem setting. Traditionally, for determining optimal execution plans, one may resort to cost-based optimization (e.g., database management systems). This paper's introduction provides precisely 1 statement to suggest that may not work for deep learning. Here's the relevant phrase: "the cost function is typically non-stationary due to the interactions between multiple devices". Unfortunately, this statement raises more questions than it answers. Why are the cost functions non-stationary? What exactly makes them dynamic? Are we talking about a multi-tenancy setting where multiple processes execute on the same device? Unlikely, because GPUs are involved. Without a proper motivation, its difficult to appreciate the methods devised.

Pros:
- Jointly optimizing forming of groups and placing these seems to have merit
- Experiments show improvements over placement by human "experts"
- Targets an important problem

Cons:
- Related work seems inadequately referenced. There exist other linear/tensor algebra engines/systems that perform such optimization including placing operations on devices in a distributed setting. This paper should at least cite those papers and qualitatively compare against those approaches. Here's one reference (others should be easy to find): "SystemML's Optimizer: Plan Generation for Large-Scale Machine Learning Programs" by Boehm et al, IEEE Data Engineering Bulletin, 2014.
- The methods are not well motivated. There are many approaches to devising optimal execution plans, e.g., rule-based, cost-based, learning-based. In particular, what makes cost-based optimization inapplicable? Also, please provide some reasoning behind your hypothesis which seems to be that while costs may be dynamic, optimally forming groups and placing them is learn-able.
- The template seems off. I don't see the usual two lines under the title ("Anonymous authors", "Paper under double-blind review").
- The title seems misleading. ".... Device Placement" seems to suggest that one is placing devices when in fact, the operators are being placed.

---

> ### Author Response · Authors · 2017-12-21
> **Response to Reviewer 3**
>
> Thank you for your constructive feedback!
>
> The reviewer is concerned with the lack of references to previous works and comparison against them. First, we are happy to add more citations to related work (see Section 1 in the updated submission). We believe that related works such as SystemML will be a strong baseline for us if we want to expand this work to memory management, since unlike runtime, memory usage is deterministic. We also compared our approach against cost-based optimization implemented in Scotch library (see Table 1) and showed that our method performs significantly better. The advantage of our method is that it’s not dependent on the hardware platform because our method can learn runtime information directly through experiments. Whereas to use Scotch, we need to feed information about the hardware platform to it.
>
> The reviewer is asking why the cost is non-stationary and dynamic, and therefore is concerned with motivation of the work. To answer this question we have added a discussion, in Section 1 on why our reward, the runtime of executing a TensorFlow graph, is non-stationary and also made it more clear that we did compare against cost-based optimizations in Table 1. In summary, in our distributed environment, we use a shared cluster of CPUs and GPUs, and our CPUs can also serve other jobs at the same time. Furthermore, in next generation of hardware platforms (such as Cloud TPUs), there will be a lot of interferences between the concurrent jobs. Again, the advantage of our method is that it’s not dependent on the hardware platform because our method can learn runtime information directly through experiments.
>
> Regarding “The template seems off. I don't see the usual two lines under the title ("Anonymous authors", "Paper under double-blind review").” and “The title seems misleading. ".... Device Placement" seems to suggest that one is placing devices when in fact, the operators are being placed.” Thanks! We fixed the formatting and will think of new names. We used device placement to be consistent with previous work.

---

> > ### Comment · AnonReviewer3 · 2018-01-11
> > **Should compare against more recent works than Scotch (circa 2009)**
> >
> > Scotch (circa 2009) seems dated. I would urge the authors to compare against more recent efforts. In case it wasn't clear from the initial review, there are other related efforts that the authors may want to compare against if they want to make the paper stronger (check the SystemML ref and/or other SystemML papers for such refs).
> >
> > As an aside, while SystemML does worry about memory (it ensures that an operators' arguments fit within device memory before placing them), it also does a number of other things such as plan enumeration, cost-based optimization, code generation, operator fusion, compression etc. etc. Describing it as a "cost-based model" that forms a "baseline for memory optimizations" does not do it justice.

---

> > > ### Author Response · Authors · 2018-01-13
> > > **RE: Should compare against more recent works than Scotch (circa 2009)**
> > >
> > > Thanks for your response!
> > >
> > > Although we cited Scotch papers from 2009, the software we used was developed in 2012: http://www.labri.fr/perso/pelegrin/scotch/. Thanks to your suggestion, we have found a more recent graph partitioning package called KaHIP, which has publications in 2017 as well as ongoing software development. We didn’t originally use it as a baseline, because unlike Scotch, it doesn’t directly address the problem of mapping a graph of operations to a graph of hardware devices. Earlier work (e.g., [1]) shows that load balancing is not always the best solution for optimizing placements. For completeness, we are now converting our graphs into a compatible format and running experiments using KaHIP and will add these results to the final version of the paper.  We are also happy to compare against new approaches suggested by reviewers.
> > >
> > > Regarding SystemML’s other optimizations, such as op fusion, compression, etc. (we also looked at its citations and other relevant papers), we want to emphasize that our paper focuses on a specific problem: partitioning operations in a computational graph so as to minimize runtime. While there are other approaches to optimizing graphs, they are complementary and/or orthogonal to our approach. We only mentioned memory optimization because we plan to apply our method to this problem in the future. Nevertheless, we looked further into references of SystemML and found a number of papers proposing cost model optimizations of computational graphs. Within those references (and references to those references), we did not find any approach that directly addresses our problem. Among those, we found [2] most relevant, which is a method that optimizes resource allocation for programs. [2] assumes a compiler that breaks the program into blocks. Given the blocks, the method searches over a grid of possible resources to be allocated to each block and finds a configuration that minimizes the cost. This method and related approaches cannot be directly applied to our problem because such a partitioner does not exist in our setting. In fact, a contribution of our work is to learn to partition a complex deep net with tens of thousands of operations into blocks and then place them to minimize the runtime. Our work shows the importance of jointly learning this partitioning together with our resource allocation over prior work such as [1] that uses a fixed set of blocks. If we were to directly apply [2] to our problem without partitioning, this would involve searching over a space of 9^80000, which is prohibitively expensive.
> > >
> > > [1] Mirhoseini A, Pham H, Le Q V, et al. Device Placement Optimization with Reinforcement Learning, ICML'17.
> > > [2] Botong Huang, Matthias Boehm, Yuanyuan Tian, Berthold Reinwald, Shirish Tatikonda, and Frederick R. Reiss. 2015. Resource Elasticity for Large-Scale Machine Learning, SIGMOD '15.

---

### Official Review · AnonReviewer1 · 2017-11-27
**In a previous work [1], an auto-placement (better model partition on multi GPUs) method was proposed to accelerate a TensorFlow model’s runtime. However, this method requires the rule-based co-locating step, in order to resolve this problem, the authors of this paper purposed a fully connect network (FCN) to replace the co-location step.**

**Rating:** 5
**Confidence:** 4

**Review:**

In a previous work [1], an auto-placement (better model partition on multi GPUs) method was proposed to accelerate a TensorFlow model’s runtime. However, this method requires the rule-based co-locating step, in order to resolve this problem, the authors of this paper purposed a fully connect network (FCN) to replace the co-location step. In particular, hand-crafted features are fed to the FCN and the output is the prediction of group id of this operation. Then all the embeddings in each group are averaged to serve as the input of a seq2seq encoder.

Overall speaking, this work is quite interesting. However, it also has several limitations, as explained below.

First, the computational cost of the proposed method seems very high. It may take more than one day on 320-640 GPUs for training (I did not find enough details in this paper, but the training complexity will be no less than the in [1]). This makes it very hard to reproduce the experimental results (in order to verify it), and its practical value becomes quite restrictive (very few organizations can afford such a cost).

Second, as the author mentioned, it’s hard to compare the experimental results in this paper wit those in [1] because different hardware devices and software versions were used. However, this is not a very sound excuse. I would encourage the authors to implement colocRL [1] on their own hardware and software systems, and make direct comparison. Otherwise, it is very hard to tell whether there is improvement, and how significant the improvement is. In addition, it would be better to have some analysis on the end-to-end runtime efficiency and the effectiveness of the placements.

 [1] Mirhoseini A, Pham H, Le Q V, et al. Device Placement Optimization with Reinforcement Learning[J]. arXiv preprint arXiv:1706.04972, 2017. https://arxiv.org/pdf/1706.04972.pdf

---

> ### Author Response · Authors · 2017-12-21
> **Response to Reviewer 1**
>
> Thank you for your constructive feedback!
>
> The reviewer is concerned that the policy training for device placement takes so many resources and quoted “320-640 GPUs” being used. In reality, we use 36 GPUs in our experiments (or 68 GPUs for deep networks). We apologize this was not clear in the paper. [More details can be found in Section 2 under “Distributed Training”.]
>
> Regarding concern about reproducibility, we confirm that it’s possible to replicate the experiments with only 5 K40 GPUs. We ran an experiment to partition 4-layer NMT model on 5 devices (1 CPU and 4 GPUs). We used 5 GPUs, 1 for policy training and 4 for measuring time, and it took roughly 2.5 hours to find a good placement. While this may seem slow, it actually takes around 12.5 GPU-hours to save 265 GPU-hours on training NMT on WMT’14 En->Fr for one epoch. [More details can be found below and in Section 3 including Fig. 3 under “Overhead of Training Hierarchical Planner”.]
>
> The reviewer is concerned with the lack of comparison against ColocRL. We want to emphasize CoLocRL makes a strong assumption that we have a human expert to manually assign operations to groups. Our method does not make this assumption. In addition to being more flexible, our method uses much fewer resources and actually gets better results. For example for NMT (2-layer), our improvement over best heuristics is 60.6%, compared to 19.0% reported in ColocRL. For NMT (4-layer) and NMT (8-layer), no results were reported for ColocRL, which we suspect is due to the model being unable to handle the large number of operations in these graphs.

---

### Official Review · AnonReviewer2 · 2017-11-27
**Elegant method with impressive results**

**Rating:** 8
**Confidence:** 5

**Review:**

This paper proposes a device placement algorithm to place operations of tensorflow on devices.

Pros:

1. It is a novel approach which trains the placement end to end.
2. The experiments are solid to demonstrate this method works very well.
3. The writing is easy to follow.
4. This would be a very useful tool for the community if open sourced.

Cons:

1. It is not very clear in the paper whether the training happens for each model yielding separate agents, or a shared agent is trained and used for all kinds of models. The latter would be more exciting. The adjacency matrix varies size for different graphs, so I guess a separate agent is trained for each graph? However, if the agent is not shared, why not just use integer to represent each operation in the graph, since overfitting would be more desirable in this case.
2. Averaging the embedding is hard to understand especially for the output sizes and number of outputs.
3. It is not clear how the adjacency information is used.

---

> ### Author Response · Authors · 2017-12-21
> **Response to Reviewer 2**
>
> Thank you for your positive feedback. We will open-source our code once the paper gets accepted.
>
> The reviewer asks if we are training a policy per model (which is the case) and whether it’s possible to use different embeddings for different ops, because it’s easier to overfit. While this is true, training the policy network will take longer without the shared embeddings. We actually tried this, and it took longer to train the policy network because the policy network has more parameters to learn.
>
> The reviewer is concerned that “averaging is hard to understand especially for the output sizes and number of outputs.” We apologize for this as this is not exactly what we did. We corrected our paper as we are not averaging the operation embeddings, but we are using information about operations assigned to a group to make a new embedding for those groups.
>
> More details are as follows which also include how adjacency information is used (we also added these details in Section 2 of the submission).
>
> First, for creating embedding for each operation, we concatenate 3 vectors:
> 1) A vector that embeds operation type information. We learn this vector similarly to how language model embedding is learned. Our vocabulary is the set of all TF operations and we learn an operation embedding of size 20.
>  2) A vector that contains output sizes and number of outputs for an operation. We set a fixed threshold (6 in our design) for maximum number of possible output edges for an operation, and for each output edge we set a threshold (4 in our design) for maximum dimension. We fill this vector of size 24 by reading the outputs of an operation one by one and putting in the output operations shapes. We fill the vector with -1 for non-existing outputs edges or dimensions.
> 3) A vector that contains adjacency information for that operation. We index the graph by traversing it in a BFS manner and set the maximum number of incoming and outgoing edges to 12 (6 for each direction). We then fill the vector with the index of the incoming and outgoing operations. We fill the vector with -1, for non-existing edges.
>
> Second, to create an embedding for each group, we concatenate 3 vectors:
> 1) A vector that counts how many of each operation types are assigned to that group. The size of this vector is the size of vocabulary of of TensorFlow’s most widely used operations which we limit to 200.
> 2) A vector that counts the overall output shapes of all the operations in that group. This vector is created by adding all the operation output shape embedding described above  (not including the -1) and is of size 16.
>  3) A vector that contains group adjacency information. The size of this vector is the number of groups (256 in our experiments), and its i'th value is 1 if the group has edges to the i'th group and is 0 otherwise.

---

> > ### Public Comment · (anonymous) · 2018-07-25
> > **code**
> >
> > Excuse me , is your code already open source?

---

### Author Response · Authors · 2018-01-05
**Summary of final revisions**

Thanks to helpful feedback from all three reviewers, we were able to significantly improve the paper!

Two reviewers have given us low scores, and we believe that the main reason for the lack of enthusiasm is that the reviewers don’t believe that device placement is worthwhile. We want to emphasize that our method saves a lot of time. For example, it takes us around 12.5 GPU-hours to save 265 GPU-hours on training NMT for one epoch on WMT’14 En->Fr. Our method finds optimized, non-trivial placements for computational graphs with over *80,000 operations*. Not only does our approach achieve strong results, but unlike previous methods which require human experts to feed in properties of the hardware or manually cluster operations, our method is end-to-end and scales to much larger computational graphs and novel hardware devices.

Based on reviewer suggestions, we made several changes to the paper, including:
  -To address Reviewer 1’s concern that the policy training for device placement “may take more than one day on 320-640 GPUs”, we’ve updated the paper to clarify that we actually use 36 GPUs (or 68 GPUs for deep networks) for at most three hours. Furthermore, we ran additional experiments in which we achieved comparable results using only 5 GPUs for 2.5 hours, which means that it takes us around 12.5 GPU-hours to save 265 GPU-hours on training NMT for one epoch on WMT’14 En->Fr. For more experiments and discussions, please see the newly added subsection called “Overhead of Training Hierarchical Planner” in Section 3.
  -To address Reviewer 3’s concern about the motivation behind our method and the non-stationarity of the reward, we have added discussions to Section 1 explaining that we use a standard cloud environment with a shared cluster of CPUs and GPUs. Therefore, our CPUs serve other jobs concurrently, making our cost function non-stationary. We also want to make it clear that we *did* compare against cost-based optimizations in Table 1 and that we achieve significantly better results.

Incorporating reviewer feedback has made our paper much stronger, so please consider updating your scores. Thanks for taking the time to review our paper!

---

### Decision · Program_Chairs · 2018-01-29
**ICLR 2018 Conference Acceptance Decision**

**Decision:**

Accept (Poster)

**Comment:**

The authors provide an alternative method to [1] for placement of ops in blocks. The results are shown to be an improvement over prior RL based placement in [1] and superior to *some* (maybe not the best) earlier methods for operations placements. The paper seems to have benefited strongly from reviewer feedback and seems like a reasonable contribution. We hope that the implementation may be made available to the community.

[1] Mirhoseini A, Pham H, Le Q V, et al. Device Placement Optimization with Reinforcement Learning[J]. arXiv preprint arXiv:1706.04972, 2017.